# Auto-Regressive Integrated Moving-Average Machine Learning for Damage Identification of Steel Frames

Yuqing Gao [1,2], Khalid M. Mosalam [1,2,*], Yueshi Chen [3,4], Wei Wang [4] and Yiyi Chen [4]

1    Department of Civil and Environmental Engineering & Pacific Earthquake Engineering Research (PEER) Center, University of California, Berkeley, CA 94720-1710, USA; gaoyuqing@berkeley.edu
2    Tsinghua-Berkeley Shenzhen Institute (TBSI), Berkeley, CA 94720-1710, USA
3    China Construction Eighth Engineering Division Co. Ltd., Shanghai 200112, China; chenyosea@gmail.com
4    State Key Laboratory of Disaster Reduction in Civil Engineering, Tongji University, Shanghai 200092, China; weiwang@tongji.edu.cn (W.W.); yiyichen@tongji.edu.cn (Y.C.)
*    Correspondence: mosalam@berkeley.edu

**Abstract:** Auto-regressive (AR) time series (TS) models are useful for structural damage detection in vibration-based structural health monitoring (SHM). However, certain limitations, e.g., non-stationarity and subjective feature selection, have reduced its wide-spread use. With increasing trends in machine learning (ML) technologies, automated structural damage recognition is becoming popular and attracting many researchers. In this paper, we combined TS modeling and ML classification to automatically extract damage features and overcome the limitation of non-stationarity. We propose a two-stage framework, namely auto-regressive integrated moving-average machine learning (ARIMA-ML) with modules for pre-processing, model parameter determination, feature extraction, and classification. Based on shaking table tests of a space steel frame, floor acceleration data were collected and labeled according to experimental observations and records. Subsequently, we designed three damage classification tasks for: (1) global damage detection, (2) local damage detection, and (3) local damage pattern recognition. The results from these three tasks indicated the robustness and accuracy of the proposed framework where 97%, 98%, and 80% average segment accuracy were achieved, respectively. The confusion matrix results showed the unbiased model performance even under an imbalanced-class distribution. In summary, the presented study revealed the high potential of the proposed ARIMA-ML framework in vibration-based SHM.

**Keywords:** auto-regressive modeling; damage identification; machine learning; steel braced frame; structural health monitoring; time series analysis





## 1. Introduction

During the lifecycle, structures are usually subjected to different types of loading stages, from service conditions to extreme events, e.g., earthquakes, tsunamis, and hurricanes, which cause different degrees and types of structural damage leading to downtimes, losses, injuries, and sometimes mortalities. Therefore, structural health monitoring (SHM) has become an important domain of research and applications in structural engineering during the last few decades. Several approaches and damage criteria have been developed for SHM, and many of them utilize vibration-based damage detection, and have received considerable attention in the context of a statistical pattern recognition paradigm [1].

This paradigm includes four major steps: (1) operational evaluation, (2) structural data collection, (3) damage-sensitive feature extraction, and (4) statistical model development for classification, where engineers use the collected signals from sensors to determine the location, type, and extent of the damage and make decisions (e.g., repair strategy) about the health status of the structures.

Damage feature extraction is an open subject in statistical pattern recognition. Many studies focus on how to locate, extract, and interpret the damage feature [2–5]. Since the

turn of this century, time series (TS) modeling of vibration signals using a family of auto-regressive (AR) models was found to be effective in damage detection and has been used to capture damage features in structures. According to Mei et al. [6], past studies using AR series models are grouped into: (1) residual-based and (2) coefficient-based models, where the former identifies damage through the residuals computed from the difference between the measured and fitted data, and the latter uses the coefficients of the fitted model as damage features.

The residual-based approach is founded on the facts that a well-fitted model trained by undamaged data is not a good fit for damaged data and that residuals increase with the increase of the damage. Sohn & Farrar [7] constructed a two-stage prediction model combining AR and AR with exogenous (ARX) models for fitting acceleration signals. They defined a damage-sensitive feature related to the standard deviation of the residual error using the AR-ARX model. Lynch et al. [8] designed a wireless unit that included implementation of the fast Fourier transform and TS modeling. In this unit, the ARX input model was used to fit the absolute acceleration and a damage feature related to the ratio of the standard deviations of the residuals of the ARX model obtained from the undamaged and damaged structural accelerations was also used.

A damage state is identified if this ratio exceeds a threshold value. Subsequently, Nair et al. [9] improved this algorithm using a normalized relative inter-story acceleration instead of a single floor acceleration, making the algorithm more robust and capable of detecting minor damage patterns. Other studies [10–13] modified the processing procedure and format of the residuals for simulated or experimental data.

The coefficient-based modeling has been shown to be more effective in many TS studies. Nair et al. [2] proposed an algorithm using an auto-regressive moving-average (ARMA) model for the vibration signals with a damage feature expressed as a function of the first three AR coefficients, and localization indices introduced based on the AR coefficients. The algorithm detected the existence of all damage patterns, including minor ones. Noh et al. [3] selected the first three regressive coefficients as the feature vector and used a Gaussian mixture model and Mahalanobis distance to detect, quantify, and localize the damage. Xing & Mita [14] proposed a substructuring approach that combined ARMA with an exogenous (ARMAX) model to identify localized damage in any story of a shear building with a limited number of sensors.

Recently, Mei et al. [6] combined both coefficient-based and residual-based approaches with the ARMAX model and used the normalized Kolmogorov–Smirnov statistical distance between two sets of ARMAX model residuals obtained from the input–output process for the undamaged and damaged states. They validated their approach on a small-scale five-story frame using aluminum floor slabs and bronze columns. *In addition to* pure TS methods, Lakshmi et al. [15] combined singular spectrum analysis (SSA) with the ARMAX model to detect minor damage, e.g., small cracks in concrete structures.

There are some drawbacks limiting the use of AR series modeling in practice. The most notable is the requirement of stationary input, which is difficult to achieve in real SHM applications, where TS data (i.e., vibration signals) collected from sensors after earthquakes are usually non-stationary. Thus, elaborate data pre-processing (e.g., segmenting, de-trending, and de-nosing) and stationarity checks are inevitable before modeling; however, these methods lack a systematic pipeline and may not guarantee stationarity. In this paper, we proposed a systematic two-stage pipeline to adopt differencing as a pre-processing approach before TS fitting to satisfy the stationarity condition. This approach is known as auto-regressive integrated moving-average (ARIMA) modeling [16]. There are very few cases in the literature that used ARIMA in SHM.

Omenzetter & Brownjohn [17] formulated a vector seasonal ARIMA model whose coefficients varied with time and were identified using an adaptive Kalman filter. Through the analysis of signals recorded during and after the construction of an instrumented bridge, they observed changes in the model coefficients related to damage under normal operational and environmental conditions. Recently, Yang & Bai [18] compared the results

between SSA and ARIMA in the problem of forecasting short-term and long-term structural strain variations.

The implementation of machine learning (ML) in SHM pattern recognition is a mature subject with well-developed and efficient algorithms [19]. ML has three major categories: unsupervised learning, supervised learning, and reinforcement learning based on data characteristics. In this paper, we only consider supervised learning, which uses well-labeled data based on domain knowledge to analyze the input data and produce an inferred mapping function [20]. The new unseen data sample can be assigned labels according to this mapping relationship. In SHM, a supervised learning algorithm learns hidden relationships between some features extracted from the data and the corresponding damaged state of the structure.

Thus, the feature extraction is an important part in applying ML in SHM where previous studies [2,3,9] designed functions for efficient damage features, e.g., combinations of coefficients of the AR models. However, the definition of such functions is based on intuitive feature engineering and is data-dependent. For more general scenarios, these functions may not be effective and the extracted features in this way may not be sufficient for the ML classification. Thus, this inspired us to consider engaging all coefficients of the AR series model as the damage features in the ML classification, which is thought to automatically maintain the information.

Most past validation experiments were based on computer simulation data, e.g., generated by the finite element method (FEM) or on reduced-scale experimental data but with pre-assigned damage patterns and locations, e.g., with component replacement or section reduction. However, during natural hazards, e.g., earthquakes, structural damage may differ from the idealization in these artificial cases. For example, refs. [3,9] used the ASCE benchmark steel frame tests where the damage patterns were artificially designed by decreasing the cross-sectional areas to simulate stiffness degradation after damage. De Lautour & Omenzetter [21] used numerically simulated three-story shear building using FEM.

Mei et al. [6] used both numerically simulated structure and laboratory test of a small-scale five-story frame where damage patterns were designed by replacing some components. In addition, there exist few studies that used the data from full-scale structures, e.g., [13,17] used full-scale instrumented bridges under normal operational and environmental conditions with minor damage levels. In summary, past applications are limited, and more realistic damage patterns of full-scale structures due to real or simulated hazards, e.g., using shaking table tests, are needed for further development of SHM using ML.

The main contribution of this study is to develop a systematic two-stage framework, namely ARIMA-ML, to combine TS modeling techniques and ML approaches for detecting structural damage. The first stage focuses on the TS modeling, and the second stage performs the recognition tasks. Specifically, ARIMA-ML consists of four main modules: (1) pre-processing, (2) model parameter determination, (3) feature extraction, and (4) classification. The performance of the framework was validated using data from full-scale shaking table tests of a three-story steel frame making use of the average segment accuracy and confusion matrix. In this study, the feature importance (*FI*) score was analyzed to examine the most important features for damage detection and pattern recognition, illustrating the need for higher order coefficients and validating the superiority of the proposed framework. The specific contributions are summarized below:

- To relax the stationarity requirement, the ARIMA model was introduced accompanied with the model parameter selection criteria and a candidate model mechanism.
- Instead of using intuition to determine a suitable function of the ARIMA coefficients as damage features, all the coefficients were used as damage features to build a $(p + q)$-dimensional vector space where $p$ and $q$ are, respectively, the number of coefficients in the AR and MA parts of the ARIMA model.
- In the classification module, an ensemble voting classifier (EVC) [22] was applied to improve both the accuracy and robustness, where multiple ML classifiers, e.g.,

support vector machine (SVM), neural network (NN), random forest (RF), and logistic regression (LR), were engaged.
• The framework was validated using realistic full-scale shaking table tests of a three-story steel frame under different earthquake hazard levels.

## 2. Time Series Modeling

### 2.1. Auto-Regressive Series

In classical TS modeling, the relationships between different variables are established through regression analyses to link different target observations. The AR series is a family of regression models used for general TS problems [16], which includes AR, moving-average (MA), ARMA, ARX, ARIMA, etc. There are several SHM applications using these models but with fewer cases of ARIMA, which is the focus of this study. We denote the basic AR model with order $p$ as AR($p$), which aims to determine the dependent relationship of an observation with the order $p$ of lagged observations. Similarly, we denote the MA model with order $q$ as MA($q$), which aims to determine the dependent relationship of the observation with order $q$ of observed white noise error terms.

$$\text{AR}(p): \quad x(t) = \sum_{i=1}^{p} \alpha_i x(t-i) + \epsilon(t) + c, \tag{1}$$

$$\text{MA}(q): \quad x(t) = \epsilon(t) + \sum_{i=1}^{q} \beta_i \epsilon(t-i), \tag{2}$$

where the TS $x$ at the current discrete time $t$ is $x(t)$, which depends on the values at the previous time steps, i.e., $x(t-i)$, $\alpha_i$ and $\beta_i$ are the coefficients corresponding to the $i$-th order term of AR($p$) and MA($q$), respectively; the white noise $\epsilon$ at time $t$ is $\epsilon(t)$; and $c$ is a constant.

### 2.2. ARMA Model in Structural Identification

Past studies revealed the coherent relationships between the ARMA parameters and structural properties in system identification [23]. Based on these efforts, a mapping between the physical parameters of a multi-degree-of-freedom (MDOF) system and the ARIMA parameters is derived herein. Based on the dynamics of structures, an $n$ MDOF system excited by a ground motion is represented by a second-order differential equation, Equation (3), where **u** is the relative displacement vector with the superposed dot indicating the time derivative, and **M**, **C**, and **K** are the $n \times n$ mass, damping, and stiffness matrices, respectively. If the input ground motion acceleration, $\ddot{u}_g(t)$ assumed to be white noise, i.e., $w(t) = \ddot{u}_g(t)$, with **1** as an $n \times 1$ unit vector, the right-hand side can be rewritten as the product of an $n \times 1$ spatial distribution vector $\mathbf{\Gamma} = -\mathbf{M1}$ as follows,

$$\mathbf{M\ddot{u}} + \mathbf{C\dot{u}} + \mathbf{Ku} = -\mathbf{M1}\ddot{u}_g(t) = \mathbf{\Gamma}w(t). \tag{3}$$

Suppose $y(t)$ represents the target single degree of freedom (DOF) response, e.g., a horizontal acceleration response of the $i$-th floor, which is a sequence of a scalar quantity. Subsequently, we convert Equation (3) to a state-space representation considering measurement noise as follows,

$$\begin{aligned} \dot{\mathbf{x}}(t) &= \mathbf{A}\mathbf{x}(t) + \mathbf{B}w(t), \\ y(t) &= \mathbf{D}\mathbf{x}(t) + v(t), \end{aligned} \tag{4}$$

where $\mathbf{x} = \begin{bmatrix} \mathbf{u} \\ \dot{\mathbf{u}} \end{bmatrix}_{2n \times 1}$ is the system state vector, $\mathbf{A} = \begin{bmatrix} [\mathbf{0}]_{n \times n} & [\mathbf{I}]_{n \times n} \\ -\mathbf{M}^{-1}\mathbf{K} & -\mathbf{M}^{-1}\mathbf{C} \end{bmatrix}_{2n \times 2n}$ is the state matrix, $[\mathbf{0}]$ is the zero matrix, $[\mathbf{I}]$ is the identity matrix, $\mathbf{B} = \begin{bmatrix} [\mathbf{0}]_{n \times n} \\ \mathbf{M}^{-1}\mathbf{\Gamma} \end{bmatrix}_{2n \times 1}$ is the input matrix, $\mathbf{D}$ is a $1 \times 2n$ output vector, and $v(t)$ is a scalar measurement noise.

Based on [23], the general solution of $\mathbf{x}$ for uniformly sampled data with the time interval $\Delta_t$ is expressed in Equation (5), where $t$ in discrete time represents the time index for the sampling of discrete quantities at $\Delta_t, 2\Delta_t, \ldots, k\Delta_t$. In the following text, $\mathbf{x}(k\Delta_t)$ is denoted as $\mathbf{x}(k)$ for short.

$$\mathbf{x}(k+1) = e^{\mathbf{A}\Delta_t}\mathbf{x}(k) + \int_{k\Delta_t}^{(k+1)\Delta_t} e^{\mathbf{A}[(k+1)\Delta_t - \tau]}\mathbf{B}w(\tau)d\tau. \tag{5}$$

Define $s = (k+1)\Delta_t - \tau$ and $ds = -d\tau$, and the second term in Equation (5) is rewritten as follows,

$$\mathbf{z}(k) = \int_0^{\Delta_t} e^{\mathbf{A}s}\mathbf{B}w((k+1)\Delta_t - s)ds. \tag{6}$$

Define the discrete version of the state matrix $\Phi_{2n \times 2n} = e^{\mathbf{A}\Delta_t}$, and its characteristic polynomial $P(\lambda)$ with the coefficients $\alpha_i$ ($\alpha_0 = 1$) is expressed as follows,

$$P(\lambda) = det(\Phi - \lambda\mathbf{I}) = \lambda^{2n} + \alpha_1\lambda^{2n-1} + \cdots + \alpha_{2n-1}\lambda + \alpha_{2n} = 0. \tag{7}$$

For $i = 0, 1, \ldots, 2n$, the outputs $y(k+i)$, from the second equation of Equations (4)–(6), are expressed as follows,

$$
\begin{aligned}
y(k) &= \mathbf{D}\mathbf{x}(k) + v(k), \\
y(k+1) &= \mathbf{D}\mathbf{x}(k+1) + v(k+1) = \mathbf{D}(\Phi\mathbf{x}(k) + \mathbf{z}(k)) + v(k+1), \\
\cdots &= \cdots, \\
y(k+2n) &= \mathbf{D}(\Phi^{2n}\mathbf{x}(k) + \Phi^{2n-1}\mathbf{z}(k) + \Phi^{2n-2}\mathbf{z}(k+1) + \cdots + \\
&\quad \Phi\mathbf{z}(k+2n-2) + \mathbf{z}(k+2n-1)) + v(k+2n).
\end{aligned} \tag{8}
$$

Multiply $y(k+i)$ by $\alpha_{2n-i}$ and, adding all terms from $i = 0$ to $2n$, it follows that,

$$y(k+2n) + \alpha_1 y(k+2n-1) + \cdots + \alpha_{2n-1}y(k+1) + \alpha_{2n}y(k) =$$
$$\mathbf{D}P(e^{\mathbf{A}\Delta_t})\mathbf{x}(k) + \vec{\beta}_1\mathbf{z}(k+2n-1) + \vec{\beta}_2\mathbf{z}(k+2n-2) + \cdots + \vec{\beta}_{2n}\mathbf{z}(k) + \sum_{i=0}^{2n}\alpha_i v(k+2n-i), \tag{9}$$

where $\vec{\beta}_i$ is a $1 \times n$ vector for a single output $y$, i.e., $\vec{\beta}_1 = \mathbf{D}$, $\vec{\beta}_2 = \mathbf{D}\Phi + \alpha_1\mathbf{D}$, $\ldots$, $\vec{\beta}_{2n} = \mathbf{D}\Phi^{2n-1} + \alpha_1\mathbf{D}\Phi^{2n-2} + \cdots + \alpha_{2n-2}\mathbf{D}\Phi + \alpha_{2n-1}\mathbf{D}$. Analogous to Equation (7), $P(e^{\mathbf{A}\Delta_t}) = P(\Phi) = \Phi^{2n} + \alpha_1\Phi^{2n-1} + \cdots + \alpha_{2n} = 0$ according to the Cayley–Hamilton theorem [24].

The dot-product between a deterministic vector $\vec{\beta}_i$ and a discrete Gaussian white noise vector process $\mathbf{z}(k+i)$ generates a sequence of random scalar quantities. Based on [23], these dot-product terms can be represented by a weighted sum of a discrete Gaussian white noise $e(t)$ with adapted scalar coefficients $\beta_j$ according to certain dependency constraints. This step is referred to as the $\beta$-trick in the sequel. In addition, the weighted sum of white noise $\sum_{i=0}^{2n}\alpha_i v(k+2n-i)$ can be replaced by $e(k+2n)$. After replacing $k+2n$ by $t$, Equation (9) can be expressed as follows,

$$y(t) + \sum_{i=1}^{2n}\alpha_i y(t-i) = e(t) + \sum_{j=1}^{2n}\beta_j e(t-j). \tag{10}$$

Re-denote $-\alpha_i$ as $\alpha_i$, and then Equation (10) can be rewritten as follows,

$$y(t) = \sum_{i=1}^{2n}\alpha_i y(t-i) + e(t) + \sum_{j=1}^{2n}\beta_j e(t-j). \tag{11}$$

Based on several assumptions and constraints [23] used in the $\beta$-trick and $y(t)$ being a stationary TS, Equation (11) shares the same form with an ARMA($2n, 2n$) model. In other

words, $y(t)$ can be estimated by $\widehat{y}(t)$ using the previous $2n$ time steps, $y(t-1), \ldots, y(t-2n)$, a random Gaussian noise sequence, $\epsilon(t)$, and a constant $c$, as follows,

$$y(t) \approx \widehat{y}(t) = \sum_{i=1}^{2n} \alpha_i y(t-i) + \epsilon(t) + \sum_{j=1}^{2n} \beta_j \epsilon(t-j) + c. \tag{12}$$

The above derivation links the arbitrary floor response $y(t)$ from structural dynamics to an ARMA model. According to [25], the eigen values $\lambda_i$ obtained from characteristic polynomial of discrete time state matrix $\Phi$, Equation (7), can be used to identify the modal frequency and damping ratio of the structure. In other words, coefficients $\alpha_i$ of the ARMA($2n, 2n$) model are related to the structural dynamic properties, which can directly contribute to the SHM [3,9,23,25].

### 2.3. Extend ARMA to ARIMA Model

In practice, the collected TS might not satisfy the stationary conditions, where the ARMA estimation in Equation (12) does not apply. However, in many cases, it is found that a TS may achieve stationarity after several differencing applications. Thus, the AR series models, e.g., ARMA, may fit an arbitrary TS after $I$ differencing. In other words, the raw TS can be estimated by $I$ times integration of the fitted TS, as reflected by "integrated" in the name of the ARIMA model. In the implementation of the ARIMA($p, I, q$) model, the raw TS is subjected to $I$ differencing (differentiation) first, and then fitted with an ARMA($p, q$) model. Suppose $x$ satisfies the stationarity condition after the $I$ differencing of $y$, i.e., $x = \nabla^I y$. Therefore, $y$ follows an ARIMA($p, I, q$) model, and $x$ follows an ARMA($p, q$) model, where, for each time step, $t$, $x(t)$ can be expressed as follows,

$$x(t) = \nabla^I y(t) = \sum_{i=1}^{p} \alpha_i x(t-i) + \epsilon(t) + \sum_{j=1}^{q} \beta_j \epsilon(t-j) + c. \tag{13}$$

Therefore, differencing can be made to Equation (11) to form an ARIMA($2n, I, 2n$) model, where the TS is stationary after the $I$ differencing. Taking one step back from $y(t)$ in Equation (11), we have

$$
\begin{aligned}
y(t-1) \;=\; & \alpha_1 y(t-2) + \alpha_2 y(t-3) + \cdots + \alpha_{2n} y(t-2n) + e(t-1) + \\
& \beta_1 e(t-2) + \beta_2 e(t-3) + \cdots + \beta_{2n} e(t-2n-1).
\end{aligned} \tag{14}
$$

Subtracting Equation (14) from Equation (11), and dividing by the time interval $\Delta_t$, we obtain

$$
\begin{aligned}
\dot{y}(t) \;=\; & \alpha_1 \dot{y}(t-1) + \alpha_2 \dot{y}(t-2) + \cdots + \alpha_{2n} \dot{y}(t-2n) + \dot{e}(t) + \\
& \beta_1 \dot{e}(t-1) + \beta_2 \dot{e}(t-2) + \cdots + \beta_{2n} \dot{e}(t-2n)
\end{aligned} \tag{15}
$$

Similarly, the $\beta$-trick can be applied to Equation (15), and the weighted noise part can be replaced by the random Gaussian noise $\epsilon$. If $\dot{y}(t)$ satisfies the stationary condition, it can be estimated by an ARMA($2n, 2n$) model, Equation (16). In other words, $y(t)$ follows an ARIMA($2n, 1, 2n$) model. Therefore, if $y(t)$ achieves stationarity with $I$ differencing, $y(t)$ can be estimated by an ARIMA($2n, I, 2n$) model.

$$\dot{y}(t) = \widehat{\dot{y}}(t) \approx \sum_{i=1}^{2n} \alpha_i \dot{y}(t-i) + \epsilon(t) + \sum_{j=1}^{2n} \beta_j \epsilon(t-j) + c. \tag{16}$$

Previous studies [2,3,23,26] showed that the coefficients of the AR part, $\alpha_i$, are more important than the coefficients of the MA part, $\beta_i$. Thus, the above ARIMA($2n, I, 2n$) model can be simplified to an ARI($2n, I$) model. Finally, the authors in [23] provided details to

estimate and map the TS coefficients to physical parameters, e.g., modal frequencies and damping ratios.

## 3. ARIMA-ML Framework

SHM for damage detection can be considered as a classification problem within the scope of supervised learning. In this study, the major contribution is to integrate both TS modeling and ML recognition to form the two-stage framework ARIMA-ML as schematically illustrated in Figure 1. Starting from stage 1: TS modeling, one arbitrary TS data is fed to the pre-processing module. Through smoothing–segmentation–normalization–differencing (SSND) operations with the *I*-th differencing and stationarity check, multiple stationary TS segments are generated. These segments are further passed to the model parameter determination module, and then a small group of candidate $(p, q)$ pairs is determined.

In stage 2: ML recognition, with filtered TS segments and one pair $(p, q)$ from the candidates, an ARMA$(p, q)$ is applied to those segments in the feature extraction module. Subsequently, the extracted features are fed into the classification module to run multiple ML classifiers with a voting mechanism for damage state identification or pattern recognition for the SHM decision-making. The candidate models are evaluated by the *average segment accuracy* (the ratio between the number of correct predicted segments and the total number of the input segments). Stage 2 is repeated for each $(p, q)$ candidate pair, and the best model is determined with the highest average segment accuracy. From the definition of the ARIMA model, fitting the *I* differentiated segments by ARMA$(p, q)$ models is equivalent to fitting an ARIMA$(p, I, q)$ model. The details of each module are described in the following subsections.

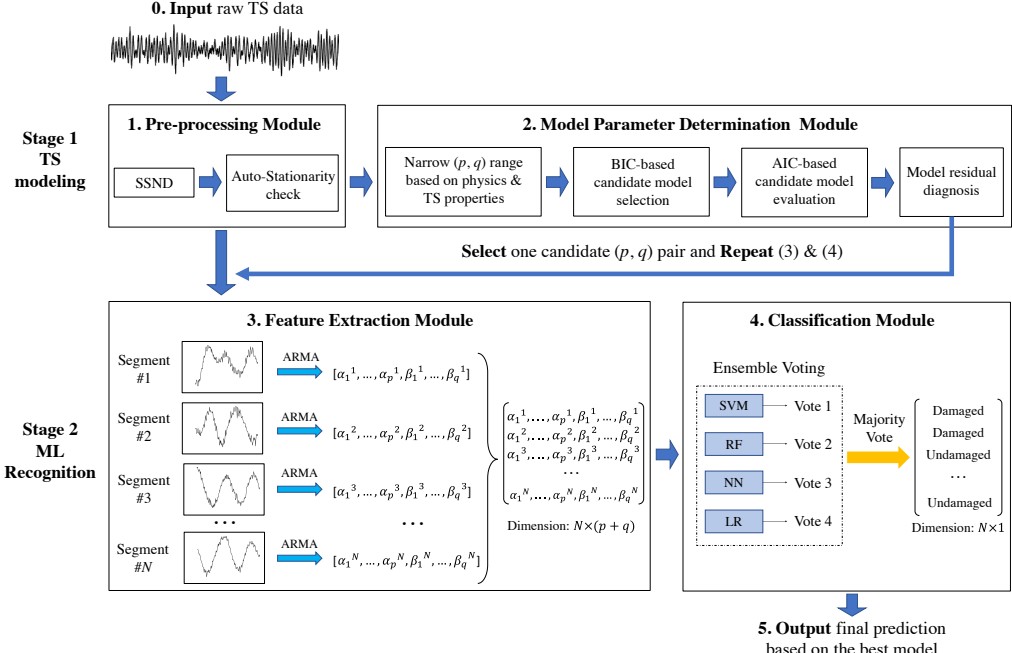

**Figure 1.** The proposed two-stage ARIMA-ML framework.

### 3.1. Pre-Processing Module

In signal processing, data pre-processing is an essential first step. In the ARIMA-ML framework, first, a moving-average operation (not to be confused with the MA model) with a suitable window size (e.g., 2 in this study) is used to smooth the raw signals and remove some noisy data points. In addition, while adopting segmentation in the TS, it is easier to achieve local stationarity for a short time period because the strong nonlinearity of the TS signal can be relieved by fitting local linear models for each segment. Moreover,

normalization (standardization) resolves the sensor-to-sensor scale variations of the TS due to different loading, calibration, or resolution conditions. Nair et al. [2] and Noh et al. [3] indicated the effectiveness of performing segmentation and normalization.

The segmented TS should maintain sufficient information of the structural properties (e.g., stiffness) where each segment contains at least one reciprocating loading cycle, e.g., the three sample segments in Figure 2. While conducting sliding window, two patterns, namely (1) *non-overlapping* (Non-OL) and (2) *overlapping* (OL), can be applied as discussed in the validation experiments. The segmentation using OL increases the amount of data analogous to data augmentation methods, which are, in general, useful for training ML algorithms, especially under limited data regimes.

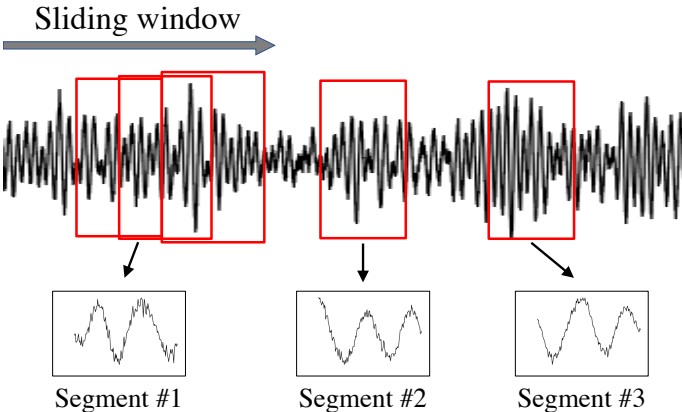

**Figure 2.** Illustration of the segmentation operation using sliding windows (overlapping (OL) for Segment #1 and Non-OL for Segments # 2 and 3) for a sample TS signal.

To minimize the variance of the local time period, instead of normalizing the whole signal, it is normalized within each segment. Let $y_j^i$ be the $j$-th segment in the $i$-th record of a collection of TS signal records, and the normalized signal, $\tilde{y}_j^i$, is obtained as follows,

$$\tilde{y}_j^i = \left(y_j^i - \mu_j^i\right)\Big/\sigma_j^i, \tag{17}$$

where $\mu_j^i$ and $\sigma_j^i$ are, respectively, the mean and standard deviation of $y_j^i$. After normalization, $I$ differencing is conducted for each segment. The selection of $I$ is based on the demand (least operations to achieve stationarity) and varies from case to case. Usually, $I = 1$ or $2$ can achieve acceptable results [16], and $I = 1$ is used in this study.

Even though SSND is conducted, stationarity of the processed data is not guaranteed. Moreover, manually examining all segments is time-consuming and impractical. Thus, an automated filtering of the non-stationary segmentation is introduced through using the Augmented Dickey–Fuller (ADF) test [27], which uses *hypothesis testing* to determine the statistical relationship in the data between two interpretations, namely the *null hypothesis* and *alternative hypothesis*. The former usually represents a commonly accepted fact or default assumption, and the latter is the opposite. They are mutually exclusive, i.e., if one is true, the other must be false.

The decision for accepting/rejecting the null hypothesis is based on whether the $p$-value (calculated probability) exceeds a designated threshold (significant level). Herein, the null hypothesis is accepted by the presence of a unit root (a stochastic trend in a TS, and its presence implies a systematic "unpredictable" pattern of the TS) in univariate TS data, which represents the non-stationarity condition. On the other hand, the alternative hypothesis is test-dependent and represents stationarity or trend-stationarity (a TS is trend-stationary if an underlying linear or nonlinear trend, e.g., a function solely given in terms of time, can be removed while leaving a stationary process [16]).

If the *p*-value from the ADF test performed after SSND on a segment is larger or equal to 0.05, the null-hypothesis is accepted, which means that the segment is non-stationary, and it is subsequently discarded. On the other hand, if the *p*-value is less than 0.05, the null-hypothesis is rejected, and the alternative hypothesis is selected, which indicates that the segment satisfies the stationarity condition.

### 3.2. Model Parameter Determination Module

Based on the physical interpretations introduced in Section 2 and reference [23], an instrumented shear-type *n*-DOF framed structural system can be modeled by fitting ARMA($2n, 2n$) or ARIMA($2n, I, 2n$) models to the collected data assuming stationary conditions and considering measurement noise. However, in practice, many other sources of noise, in addition to the measurement errors, may influence the accuracy of the model fitting, e.g., installation errors and model idealization errors. Thus, the values of *p* and *q* can be determined in a statistical manner, i.e., considered as being close to $2n$ but varied based on the observations of autocorrelation function (ACF) and partial autocorrelation function (PACF) making use of the TS properties. Shumway & Stoffer [16] stated the following about the ACF and PACF for the model selection:

- A TS suitable for the AR(*p*) model is observed after a time lag *p* as ACF tails off (ACF values gradually decrease with the lag increase) and PACF cuts off (most of the PACF values are within a certain range, e.g., bounds of the PACF confidence intervals, with the lag increase).
- A TS suitable for the MA(*q*) model is observed as ACF cuts off and PACF tails off after lag *q*.
- A TS suitable for the ARMA($p, q$) model is observed as both ACF and PACF tails off after lags *p* and *q*.

Therefore, a group of ($p, q$) pairs can be determined based on the above observations, where values of *p* and *q* are close to $2n$, or $q = 0$ if only applying the AR/ARI model.

To further narrow the selection of ($p, q$) pairs, *Bayesian information criterion* (BIC) sub-model analysis is conducted. The BIC values, from Equation (18), are computed for sub-models of ARMA($p_{max}, q_{max}$) with partial orders, which use the combinations of the orders from 1 to the highest orders $p_{max}$ and $q_{max}$ in the ($p, q$) pairs. The results demonstrate the contribution from each order, and then a small group of candidate models are selected. Subsequently, *Akaike information criterion* (AIC) analysis is performed, where average AIC values, Equation (19), are computed for each candidate model with all segments. These average AIC values evaluate the overall fitting performance of each candidate model.

$$BIC = k \ln(n) - 2 \ln(\widehat{L}), \tag{18}$$

$$AIC = 2k - 2 \ln(\widehat{L}), \tag{19}$$

where *n* is the number of samples, *k* is the number of coefficients in the model, and $\widehat{L}$ is the maximum value of the likelihood function for the model, referring to [16]. In BIC and AIC analyses, the lower (algebraically) the value, the better the fit.

Models using different ($p, q$) pairs are diagnosed by the residuals to avoid violating the assumptions of the AR series models. The residuals, computed from differences between ground truth and fitted data, should share similar properties to white noise, i.e., *independent and identical distribution* (i.i.d.), a zero mean, and a limited value of the standardized (STD) residuals (e.g., less than 2 as suggested in [28]). Statistical methods to examine the residuals include standardization plots, quantile–quantile (QQ) plots for normality, and ACF plots for autocorrelation.

However, for processing a large amount of data, manually checking each TS segment with these plots is inefficient, especially if there are many TS data to analyze as in the case of a large instrumented structural system, e.g., a continuously monitored long span bridge with many sensors. Therefore, *Ljung–Box statistics* [29] is applied herein to automatically diagnose the residuals based on the hypothesis testing [30]. Once the null hypothesis is

accepted with $p$-value exceeding the threshold of 0.05, the residuals check is satisfied. This module is summarized as follows:

1. Select multiple $(p, q)$ pairs based on physical interpretations and TS properties, e.g., the initial value of $p$ is considered close to $2n$.
2. Conduct BIC sub-model analysis to determine a small group of $(p, q)$ candidate models.
3. Conduct AIC analysis to evaluate the fitting performance of each $(p, q)$ pair.
4. Conduct residual diagnosis for each $(p, q)$ pair.

Having more candidate models may lead to a more robust model performance; however, this is more costly from a computational point of view. Empirically, we suggest to initially consider three to five candidate $(p, q)$ pairs.

### 3.3. Feature Extraction Module

The general ML classification process is described as extracting features from the raw data, determining the mapping relationship between the features and their labels, and finally tuning the models to obtain the best mapping function. Nair et al. [2] and Noh et al. [3] presented promising detection results of feature extraction using combinations of the first several AR coefficients of ARMA model. However, such features may not be sufficient for general structural systems, where damage patterns are not simply represented by reduced cross-sections as conducted in many idealized laboratory experiments.

It is known that the AR coefficients contain information about the modal frequencies and damping ratios [26], and the changes (indicative of damage) in the stiffness of the structure are accordingly reflected by the changes in these coefficients [2], which can be used for damage detection. However, from Sections 2.2 and 2.3, if an ARMA/ARIMA is used as a TS model, the MA parts may also have certain contributions. Therefore, in this module, all coefficients of the AR, $\alpha_i$ ($i \in \{1, 2, \ldots, p\}$), and MA, $\beta_j$ ($j \in \{1, 2, \ldots, q\}$), from the ARIMA model are collected and concatenated into a $(p + q)$-dimensional feature vector where several ML classifiers are applied on the whole feature vector.

If only applying a simpler AR/ARI model without considering the MA part, all $\alpha_i$ coefficients form a $p$-dimensional feature vector. In addition, since a fixed $(p, q)$ pair is applied to all TS segments, some segments may still not be well-fitted. Thus, similar to the model parameter determination module, a residual check using Ljung–Box statistics is conducted after model fitting, and the segments violating the acceptable residual assumptions are discarded.

### 3.4. Classification Module

With the extracted features, four general ML classifiers, namely SVM, NN, RF, and LR, are jointly applied for the damage classification purposes. To improve the generalization and robustness over a single classifier, an ensemble voting classifier (EVC) is adopted. The characteristics of the EVC can be described as: (i) the final predicted results are the majority of the predictions from all individual classifiers, and (ii) the results are selected based on the "best" (highest test accuracy and smaller variance if applicable) individual classifier performance in the cases of a tie in voting. For more details about these four general classifiers, refer to [22].

### 3.5. ARIMA-ML in Structural Damage Identification

Integrating the above-mentioned four modules describes the two-stage framework of the ARIMA-ML. Stage 1 is used to determine a set of TS models with multiple candidate $(p, q)$ pairs, and stage 2 is used to train the classification models. Repeating stage 2 for each candidate pair, the best TS model with its corresponding trained classification model is obtained.

While applying the ARIMA-ML framework to the structural damage identification, its input is the acceleration response collected **after** a possible damaging loading scenario, e.g., due to an earthquake. This response can be excited from small white noise signals applied to the structure from ambient vibration or from a small external force, e.g., a

hammer impact excitation as used in [31]. The underlying assumption is that the structural damage state or pattern is determined from the damaging loading stage, and the white noise excitation with low amplitude from ambient vibration or external force does not cause further damage, i.e., the structure responds linearly with a degraded stiffness compared to its undamaged state. Based on damage criteria, these TS data are labeled accordingly.

From the ARIMA-ML framework with labeled data, the best TS model, i.e., ARIMA($p, I, q$), can be obtained, and the corresponding classification models are trained. From the physical interpretation in Section 2, the order of the TS model is more related to the number of DOFs and varies according to the TS properties. Even though TS data with different damage states may differ in properties leading to different ($p, q$) selection, it is suggested to use the higher order among multiple model choices [16].

The well-trained ARIMA-ML framework can be applied automatically for real-time SHM. Through inputting newly received TS signal from the sensors, the framework automatically processes the TS data through pre-processing, feature extraction, and classification modules once, and outputs the corresponding damage state. This workflow (refer to Figure 3 for the pseudo-code of the trained ARIMA-ML framework) indicates an end-to-end efficient run-time performance, which is suitable for developing embedded intelligent devices for real-time automatic damage detection. The code implementation is slightly different from Figure 1 where the SSND is followed by feature extraction segment by segment to form a stacked feature matrix, which is then fed as vectorized data into the ML classification to accelerate the computation and increase its efficiency.

---

*Require*: time series $Y \in \{Y_1, Y_2, ..., Y_n\}$, ARIMA order ($p, I, q$), segmentation size $L$, overlapping length $OL$, trained
　　　classifiers $clf \in \{SVM, RF, NN, LR\}$
*Define*: feature matrix $F$, damage state $DS$

---

Set j = 1 # *Count for number of segmentations*
**for** $i \leftarrow 1$ to $n$ **do**
　　**Initialize** $index_{start}$ and $index_{end}$
　　**while** $index_{end} \leq length(Y_i)$ **do**
　　　　**Set** $index_{end} = index_{start} + L$
　　　　$Dseg \leftarrow SSND(Y_i[index_{start}: index_{end}],$ order $= I)$ # *SSND with differencing order I*
　　　　$p$-value $\leftarrow ADF(Dseg)$ # *Auto-stationarity Check*
　　　　**if** $p$-value $\geq 0.05$ **do** # *Check if violate stationary condition*
　　　　　　**Break** to outer loop and **Update** $index_{start} = index_{start} + OL$
　　　　**end if**
　　　　$[Coef_{AR}, Coef_{MA}], Residual \leftarrow ARMA(p, q)$ # *Feature extraction*
　　　　$p$-value $\leftarrow$ Ljung-Box statistic $(Residual)$ # *Residual check*
　　　　**if** $p$-value $\leq 0.05$ **do** # *Check if violate residual assumption*
　　　　　　**Break** to outer loop and **Update** $index_{start} = index_{start} + OL$
　　　　**end if**
　　　　**Stack** $[Coef_{AR}, Coef_{MA}]$ as new row of feature matrix $F$, $F_j$
　　　　**Update** $index_{start} = index_{start} + OL, j = j + 1$
　　**end while**
**end for**
# Classification
**Load** $Clf_1 \leftarrow SVM, Clf_2 \leftarrow RF, Clf_3 \leftarrow NN, Clf_4 \leftarrow LR$
$DS \leftarrow EVC.predict(data = F, clf = (Clf_1, Clf_2, Clf_3, Clf_4))$
**Output** $DS$

---

**Figure 3.** Pseudo-code of the trained ARIMA-ML framework run-time.

## 4. Full-Scale Shaking Table Tests

Full-scale shaking table tests were conducted on a three-story, three-bay, tension-only concentrically braced beam-through frame (TCBBF) (Figure 4) under different earthquake hazard levels. The test structure has a total height of 10,020 mm (story height = 3340 mm). Complete details about the test steel frame can be found in [32]. The acceleration sensors, Figure 5a, were installed at the center and two corners of every floor to measure the response in two horizontal directions, Figure 5b.

The structure was shaken using natural and artificial ground motions for frequently occurring earthquakes (FOE), design basis earthquakes (DBE), and the maximum considered earthquake (MCE). These levels were defined using the peak ground acceleration (PGA) according to the seismic intensity 7 in the Chinese code for the seismic design of buildings [33], namely 0.035 g, 0.1 g, and 0.22 g, respectively, where *g* is the acceleration of gravity. To evaluate the structural damage, low amplitude white noise signals, which do not cause further structural damage, were applied after each earthquake loading, and the damage patterns were subsequently recorded.

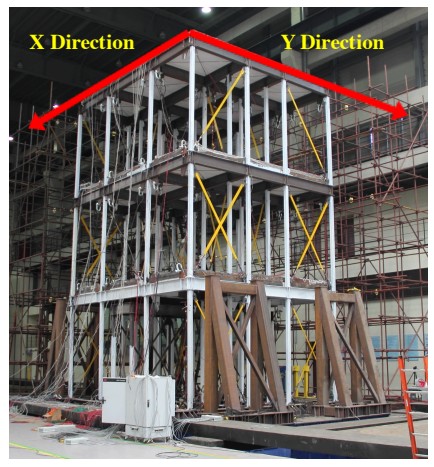

(**a**) General view

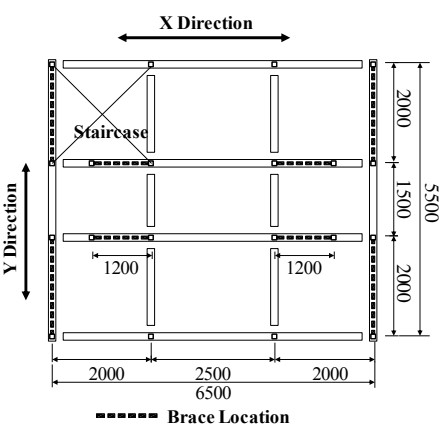

(**b**) Plan layout (Units: mm)

**Figure 4.** Three-story tension-only concentrically braced beam-through frame (TCBBF).

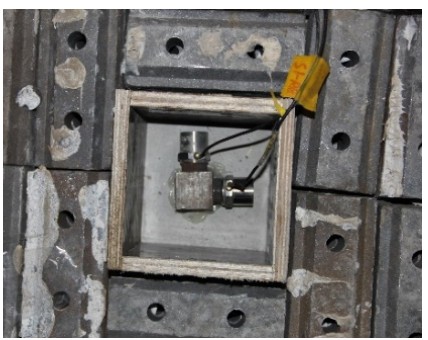

(**a**) 2D accelerometers

(**b**) Typical floor arrangement

**Figure 5.** Sensors layout.

During the tests, no fracture occurred in the structural members, i.e., the beams, columns, and beam–column joints. Therefore, the steel frame remained elastic during the loading cases. Only the braces yielded and loosened after the earthquake loading, Figure 6. Hence, there were two damage patterns observed for each floor: (1) partial braces loosened (PL), and (2) all braces loosened (AL). Both patterns were well-recorded and used for data labeling for the detection tasks, refer to Table 1. If no damage pattern was observed, it was recorded as undamaged (UD).

In the validation experiments, the used data are signals of structural acceleration response under white noise collected from the sensors on each floor. For simplicity, only the X direction (Figure 4) data were used, as more significant damage was observed in this direction.

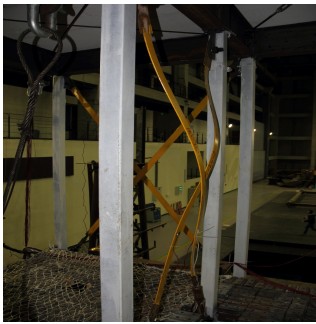
(**a**) Braces in the X direction

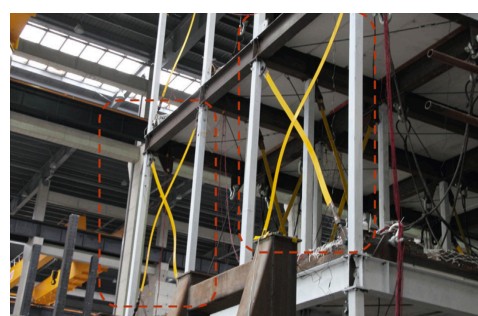
(**b**) Braces in the Y direction

**Figure 6.** Loosening of the braces.

**Table 1.** Structural information and damage patterns recorded under earthquake loading (UD: undamaged, PL: partial braces loosened, and AL: all braces loosened).

| Hazard | Record | | PGA (g) | | Fundamental | Damage Pattern |
|---|---|---|---|---|---|---|
| Level | No. | Name | X | Y | Period (sec) | |
| B-FOE [1] | EQ1 | El Centro-X [2] | 0.035 | 0.03 | 1.36 | AL |
| | EQ2 | El Centro-Y | 0.03 | 0.035 | 1.36 | AL |
| FOE | EQ3 | SHW2-X [3] | 0.035 | 0.03 | 0.56 | UD |
| | EQ4 | Kobe-X [4] | 0.035 | 0.03 | 0.56 | UD |
| | EQ5 | Kobe-Y | 0.03 | 0.035 | 0.56 | UD |
| DBE | EQ6 | El Centro-X | 0.1 | 0.085 | 0.56 | UD |
| | EQ7 | El Centro-Y | 0.085 | 0.1 | 0.56 | UD |
| | EQ8 | SHW2-X | 0.1 | - | 0.56 | UD |
| | EQ9 | Kobe-Y | 0.085 | 0.1 | 0.67 | UD in floors 1 & 2; AL in floor 3 |
| MCE | EQ10 | El Centro-X | 0.22 | 0.187 | 0.74 | PL in floors 1 & 2; AL in floor 3 |
| | EQ11 | El Centro-Y | 0.187 | 0.22 | 0.81 | PL in floors 1 & 2; AL in floor 3 |
| | EQ12 | SHW2-X | 0.22 | - | 1.13 | PL in floors 1 & 2; AL in floor 3 |
| | EQ13 | SHW2-Y | - | 0.22 | 1.13 | PL in floors 1 & 2; AL in floor 3 |
| | EQ14 | Kobe-X | 0.22 | 0.187 | 1.36 | AL in all floors |
| | EQ15 | Kobe-Y | 0.187 | 0.22 | 1.36 | AL in all floors |
| A-DBE [5] | EQ16 | El Centro-X | 0.1 | 0.085 | 0.56 | UD |
| | EQ17 | El Centro-Y | 0.085 | 0.1 | 0.56 | UD |

[1] The FOE level earthquake input before brace installation. [2] El Centro record from the 1940 California Imperial Valley earthquake. [3] Shanghai artificial accelerogram [33]. [4] JMA Kobe record from the 1995 Kobe earthquake. [5] Aftershock at the DBE level applied after re-tightening/replacing all loosened braces.

## 5. Validation Experiments

### 5.1. Experimental Objectives

Damage detection and localization are of major interests in SHM. Three tasks are pursued herein: (1) global damage detection, (2) local damage detection, and (3) local damage pattern recognition.

In Task 1, the global health condition of the whole structure was assumed to be independent of the damage location. In other words, the occurrence of any damage in the structure denoted a damaged state (a binary classification problem). For this purpose, all signals were mixed together with labels of either "undamaged" (UD) or "damaged" (D) ignoring the floor information. This can be considered as one idealization of the structure as two independent single-degree-of-freedom (SDOF) systems, one in the X direction and one in the Y direction, Figure 7.

In Task 2, a more fine-grained detection was conducted to locate the damage based on the floor level. Considering only the X direction and the rigid diaphragm assumption [34,35], the steel structure can be idealized as an MDOF system with one translational

DOF for each floor, Figure 7, and the detection accuracy was restricted to the floor level. Only signals collected from the same floor were mixed along with their damage state, and the damage detection was performed floor by floor. In other words, sensors in different locations of one floor shared the same label. Therefore, this task is three pairs of a binary classification problem.

In Task 3, the development of damage in the test steel frame was further investigated. As mentioned above, three different patterns based on the floor level were assigned: UD, PL, and AL, which is a three-class classification problem. The design of the test frame had a single brace in each of the two exterior bays (Y-direction) of floor 3, Figure 4a, and the section area of each of these braces on floor 3 was smaller than that used on the other floors, which led to its fragility under earthquake excitation. As a result, floor 3 was labeled UD or AL since it did not experience PL in the tests, Table 1. Thus, only data collected from floors 1 and 2 were evaluated in this task.

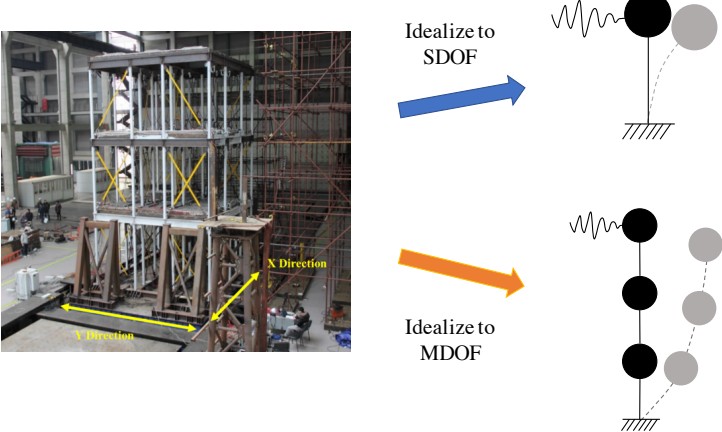

**Figure 7.** Structural idealization of the test steel structure.

In the above three tasks, the average segment accuracy was considered as the evaluation metric. To investigate the possible existence of biased prediction of each model, i.e., obtain a high accuracy by simply predicting one class (the majority) and missing the others (the minority) in an unbalanced dataset, confusion matrix analysis [22] was conducted. In addition, the importance of each coefficient was further evaluated using the feature importance (*FI*) score. While the RF classifier was performing classification, the fed-in data are split into multiple subsets, where each subset belongs to one specific class (e.g., UD, PL, or AL in our case).

The split criteria are based on the features and their weights. The more important features make the subset able to distinguish the class better, and subsequently the corresponding weights are used to compute the *FI* score. In other words, the *FI* score is a good indicator for the evaluation and better understanding of how the RF conducts the classification via features. The higher the value of the *FI* score, the more important the feature is. For more details about the *FI* score computation, refer to [36]. Moreover, since all the ARIMA coefficients are concatenated as features, the *FI* score provides guidance for the importance of each coefficient in the classification.

### 5.2. Experimental Setups

In the pre-processing module, according to the sampling frequency (256 Hz) of the sensors used in the shaking table experiments, the segmentation window size was taken as 200 data points. Thus, the duration of each segment was about 0.78 s ($1/256 \times 200 \approx 0.78$), which was considered sufficient in this study. To further investigate the influence of overlapping in the sliding window, both Non-OL and OL patterns were considered for each model. From the selected segmentation window size of 200, the OL size was assigned as 100 for convenience.

To evaluate the classification performance in ML, the collected data were separated into training and test datasets. The TS data collected after the arbitrarily selected shaking table runs EQ4, EQ7, EQ9, and EQ12 (Table 1) were used for testing, which covers the scenarios of FOE, DBE, and MCE level earthquakes, and the remaining data were used for training.

As mentioned above, there were three accelerometers installed on each floor; thus, a total of 117 (13 records × 3 floors × 3 sensor locations) TS signals and 36 (4 records × 3 floors × 3 sensor locations) TS signals were used for training and testing, respectively. To indicate the generalization of the predictive classifiers, only the test accuracy is presented in the following sections. For the feature importance analysis, the range of $FI$ scores for each feature, $FI_i$ ($i \in \{1, 2, \ldots, (p+q)\}$) was normalized to be $0.0 \leq \widetilde{FI}_i \leq 1.0$, where $\widetilde{FI}_i = FI_i \big/ \sum_{j=1}^{p+q} FI_j$, and all $\widetilde{FI}_i$ values sum up to 1.0.

To avoid loss of generality, $k$-fold cross-validation was applied and used for tuning the hyper-parameters of the classifiers. In this cross-validation, all training data were randomly and equally split into $k$ folds; at each time, $k-1$ folds with a new combination of folds were used as a training set; and the remaining $k$-th fold was assigned as a validation set until each fold had been validated once. In the following experiments, $k = 5$ was used. Multiple runs in cross-validation determined the relatively optimal setting for each classifier, Table 2.

**Table 2.** Implementation details of the classifiers.

| Classifier | Details |
|---|---|
| SVM | 1. Use kernel-SVM with Gaussian kernel. <br> 2. Use L2 regularization parameter [22] $C = 100$. |
| NN | 1. Build a two-layer network. <br> 2. Neurons in first and second layers are 200 and 100, respectively <br> 3. Use ReLU [37] as an activation function. |
| RF | 1. Use Gini impurity [36] as a split criterion. <br> 2. Use 250 trees as the estimator and then average their predictions. |
| LR | 1. Add L2 regularization term. <br> 2. Use L2 regularization parameter $C = 100$. |
| EVC | 1. Ensemble of the above four individual classifiers with their optimal settings. <br> 2. Use a hard-voting mechanism. |

## 6. Stage 1: TS Modeling

### 6.1. TS Data Pre-Processing

As the first step, all raw TS data were passed to the pre-processing module, which generated a group of stationary segments. Herein, one TS sample is presented to indicate the necessity of conducting the differencing operation and procedure of the auto-stationarity check. First, the ACF and PACF plots of the TS example are plotted in Figure 8. If only SSN is performed, Figure 8a shows certain trends in the ACF plot, and its values diminish slowly, indicating non-stationarity. By performing differencing once after SSN, the performance in both the ACF and PACF plots significantly improved, where the ACF diminished rapidly within the shown dashed lines with less than 15 lags and the PACF cut off a certain number of lags, Figure 8b. This observation roughly meets the property of the stationary condition and indicates that the differencing order $I = 1$ is sufficient.

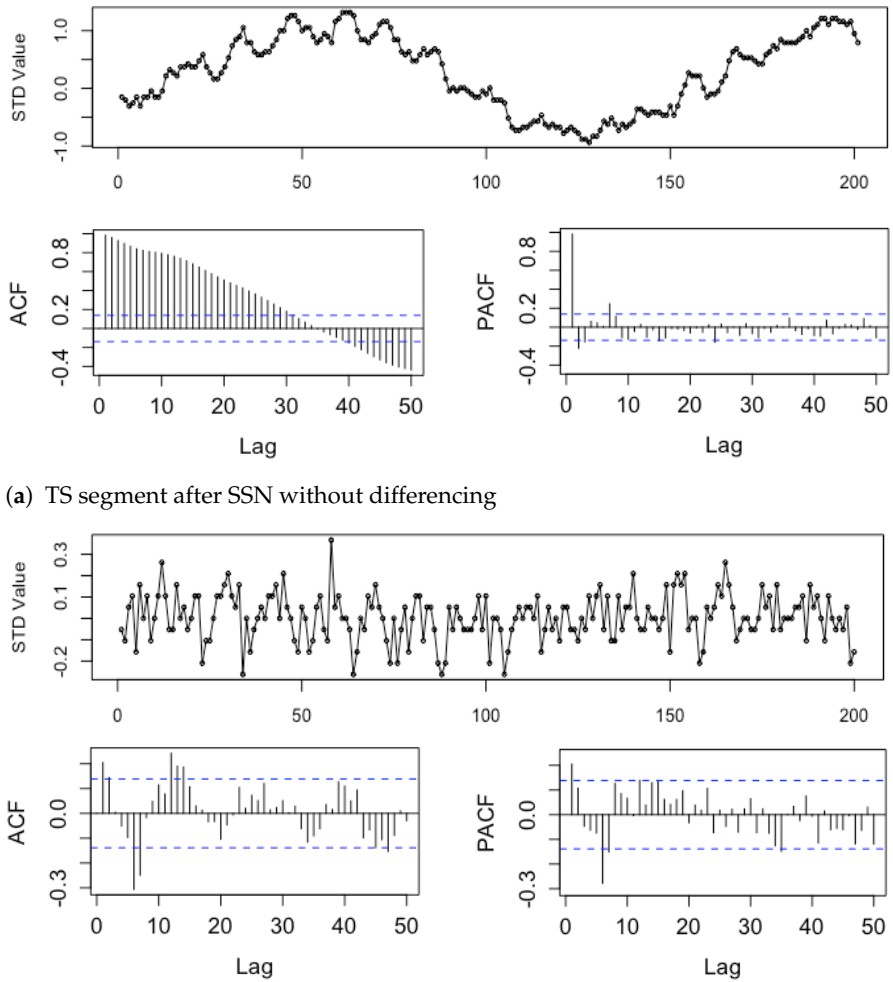

(**a**) TS segment after SSN without differencing

(**b**) TS segment after SSND with one-differencing

**Figure 8.** Standardized (STD) value, ACF, and PACF of one sample TS segment with different pre-processing. (Dashed lines: approximate 95% confidence intervals of the white noise results; vertical lines: values beyond which autocorrelations are (statistically) significantly different from zero [38]).

Subsequently, the stationarity of the TS with the ADF test was examined, where *p*-values of 0.78 and 0.01 were achieved for the cases of SSN and SSND, respectively. The former significantly exceeds 0.05, which accepts the null hypothesis (non-stationary), but the latter rejects the null hypothesis and accepts the alternative hypothesis (stationarity). After examining several plots of different segments, we concluded that stationarity was difficult to achieve for the collected data from sensors if only SSN was used. However, one-time differencing significantly resolved this issue, supporting the need of ARIMA($p, 1, q$) models. Herein, if one-time differencing still cannot satisfy the stationary condition, the segment would be discarded. Finally, the label distribution among all segments is shown in Figure 9, which is somewhat imbalanced, e.g., UD segments are about twice those of AL. It addition, both training and test datasets shared the same distribution after pre-processing.

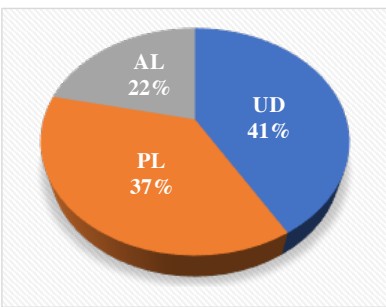

**Figure 9.** The label distribution among all segments.

### 6.2. Model Parameter Determination

#### 6.2.1. TS Properties for the Order Range Determination

The optimal orders of $p$ and $q$ for the ARIMA model or $p$ for the ARI model without the MA part were close to $2n$, i.e., 6 in this study. Subsequently, few choices were generated based on observations from the ACF and PACF plots of the TS segments. The same results of the TS example in Figure 8b are used in the following discussions. From the PACF plot, we concluded that lags 6 or 7 were the cut lag for this segment. From the ACF plot, autocorrelation diminished after lag 6 indicating that it was a tail-off lag. When generalizing to all segments, the order $p$ for the AR part can be taken as 5, 6, or 7, and the order $q$ for the MA part can be taken as 5 or 6 (0 for the ARI model). Thus, a group of models, e.g., ARI(5,1), ARI(6,1), ARI(7,1), ARIMA(5,1,6), ARIMA(6,1,6), and ARIMA(7,1,6) with $p_{max} = 7$ and $q_{max} = 6$ were initially considered.

#### 6.2.2. BIC-Based Candidate Model Selection

BIC values were explored for sub-models of ARIMA($p_{max}, 1, q_{max}$) with different combinations of $p = 0$ to 7 and $q = 0$ to 6. In Figure 10, each row represents one specific sub-model with different combinations of the model parameters for the AR($p$) and MA($q$) terms, represented by the gray color depth in each column. The horizontal axis labels the TS $x$-lag $i$, $x(t - i)$, and error-lag $j$, $\epsilon(t - j)$, as expressed in Equation (13), contributing to order $i$ AR($\alpha_i$) and order $j$ MA($\beta_j$) terms, respectively. The intercept is the bias term in the regression.

The vertical axis is the BIC computed for each sub-model. The darker the color and the lower the BIC value, the better the model. For example, in Figure 10a, the first row represents the sub-model of ARIMA(7,1,6) with the BIC value $-590$, and its parameters are related to TS $x$-lag (AR terms) 1, 2, 3, and 7 and error-lag (MA terms) 2, 4, and 6. Better models are in the top rows with lower BIC values. There were multiple sub-models sharing the same BIC values, i.e., the top four rows in Figure 10a.

Although they show contributions from different combinations of the lower order lags, the higher orders TS $x$-lag 7 and, to a lesser extent, the error-lag 6 are present in most cases. Based on the results of other segments, Figure 10b,c, the influence of TS $x$-lag 7 and error-lags 2 and 6 in dominating the BIC performance was detected. This is evidence to incorporate higher orders in the ARIMA models. Thus, combining the observations that PACF cuts off from lag 6 or 7 and ACF tails off simultaneously, ARI(7,1), ARIMA(6,1,6), and ARIMA(7,1,6) became candidate models.

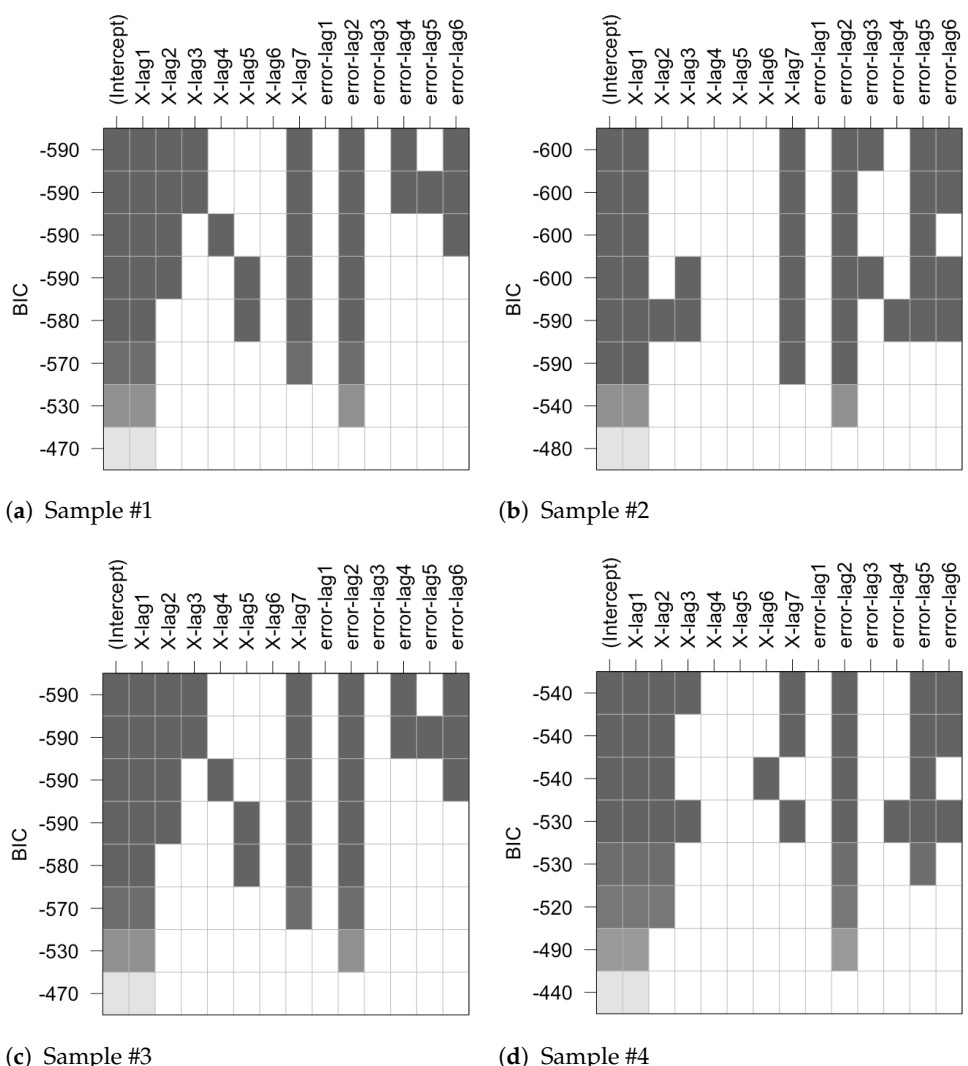

**Figure 10.** BIC values for the sub-models computed on four sample TS segments.

### 6.2.3. AIC-Based Candidate Model Evaluation

The AIC values were examined for more insight into the candidate models. Since segmentation operation was conducted, which caused the AIC value to vary from one segment to another, the average AIC values were computed from all segments of each model. In addition, the datasets were split into training and test sets and the average AIC values of each model were computed for both sets. To explore the influence of overlapping in the SSND for data augmentation, the average AIC values were also evaluated for both Non-OL and OL patterns. However, OL was only performed on the training data.

Table 3 shows that a sliding window with OL slightly improved the average AIC, which demonstrates the effectiveness of using OL for data augmentation. In addition, the values from the test set were close and even slightly better than those from the training set. This observation indicates the stable fitting performance and the generalization of these TS models. It is also shown that the ARIMA(6,1,6) and the ARIMA(7,1,6) achieved similar average AIC values with the latter slightly lower (algebraically), and both models had significantly lower values than that of the simpler ARI(7,1) model. This observation reveals that more complex models fit the data better than simpler ones.

**Table 3.** Average AIC of the three candidate models.

| Set | Pattern | ARI(7,1) | ARIMA(6,1,6) | ARIMA(7,1,6) |
|---|---|---|---|---|
| Training | Non-OL | −417.11 | −442.54 | −448.97 |
| | OL | −444.50 | −470.51 | −477.09 |
| Test | Non-OL | −431.11 | −459.06 | −464.14 |

### 6.2.4. Model Residual Diagnosis

After fitting, the residual diagnosis was performed. From the results, the three models achieved similar performances, Figure 11. Only a few outliers fell outside the two STD residuals, and most points were within one STD residuals, indicating a normal distribution. The correlation of the residuals using the ACF diminished and remained bounded by the white noise bounds (dash lines), thus, indicating a low correlation between time lags of the residuals. From the QQ-plot, most points aligned along one line, which implies i.i.d. Hence, all three models satisfied the residual assumptions where the ARIMA(6,1,6) and ARIMA(7,1,6) models were slightly superior, compared with the ARI(7,1) model, with fewer outliers in the ACF, STD residuals, and QQ-plots.

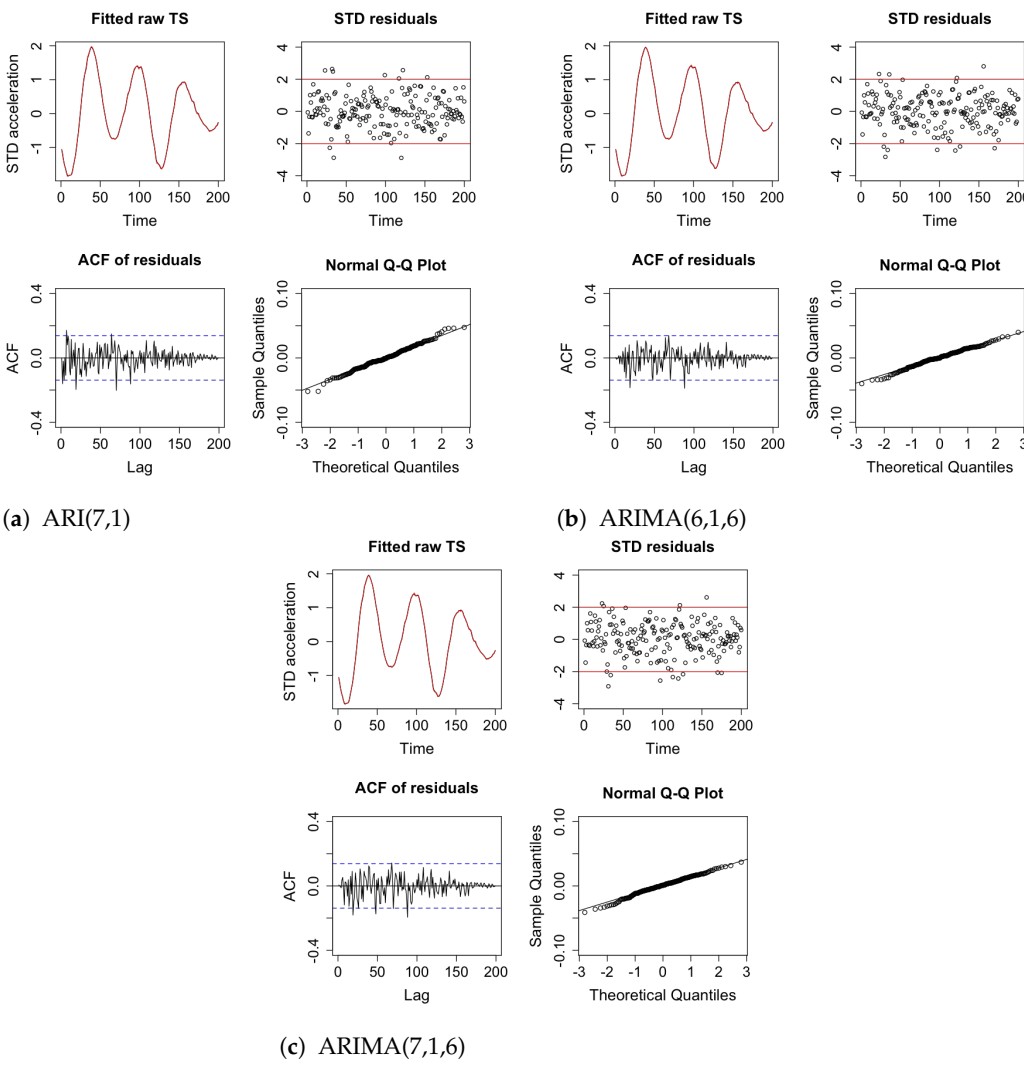

**Figure 11.** Residual diagnosis for the three candidate models.

## 7. Stage 2: ML Recognition

In stage 1, raw TS data were pre-processed to multiple segments, and then ARI(7,1), ARIMA(6,1,6), and ARIMA(7,1,6) were determined as candidate models with stable and generalized performance satisfying residual assumptions. In stage 2, the segments were further fed into each TS model to extract features for classification, and finally the best model was selected based on the highest average segment accuracy. Some segments were discarded if the residuals violated the i.i.d. assumption.

Accordingly, different models generated different numbers of feature–label pairs due to different model complexity and constraints, e.g., the relatively simple ARI(7,1) model had more data due to less modeling constraints. In addition, the feature–label pairs generated by OL were about twice those generated by Non-OL, refer to Table 4. In general, the amount of feature–label pairs of the three models was close and did not influence the performance comparison between these models.

**Table 4.** The number of feature–label pairs generated by different models.

| Dataset | Pattern | ARI(7,1) | ARIMA(6,1,6) | ARIMA(7,1,6) |
|---------|---------|----------|--------------|--------------|
| Training | Non-OL | 7396 | 7155 | 7132 |
| | OL | 14,771 | 14,287 | 14,273 |
| Test | Non-OL | 2559 | 2488 | 2471 |

### 7.1. Task 1: Global Damage Detection

The results of Task 1 are presented in Table 5. There was no significant difference in the accuracy between Non-OL and OL, ≤0.3% in EVC. Accordingly, the generated feature–label pairs without OL ($\approx$7000) contained sufficient information to accurately train a classifier for global damage detection. Therefore, the ARIMA-ML is proposed to be applicable in practice since it does not require a very large dataset. In addition, due to the similar performance of Non-OL and OL and for computational efficiency, only the Non-OL pattern was considered in Tasks 2 and 3.

**Table 5.** Accuracy (%) of the global damage detection for different candidate models. Ensemble voting classifier (EVC).

| Model | Pattern | SVM | NN | RF | LR | EVC |
|-------|---------|-----|-----|-----|-----|-----|
| ARI(7,1) | Non-OL | 97.2 | 97.5 | 96.4 | 97.0 | 97.6 |
| | OL | 97.3 | 97.5 | 96.8 | 97.2 | 97.7 |
| ARIMA(6,1,6) | Non-OL | 96.4 | 96.7 | 91.3 | 93.4 | 96.7 |
| | OL | 96.6 | 96.9 | 92.5 | 93.5 | 96.9 |
| ARIMA(7,1,6) | Non-OL | 96.6 | 96.3 | 92.1 | 93.5 | 96.6 |
| | OL | 96.9 | 96.4 | 92.9 | 93.7 | 96.9 |

Among different individual ML classifiers, SVM and NN classifiers achieved the highest accuracy, while RF and LR were slightly lower. Moreover, the ensemble classifier EVC achieved a stable and robust performance, comparable to the best individual classifier [36,39]. This confirmed the effectiveness of using ensemble learning in this study. In general, the best classification performance using features extracted from these candidate models were similar (96.6~97.7%), and the EVC classifier using features extracted by ARI(7,1) was slightly better than the other two, i.e., the best fitting model may not be best feature extractor. This is mainly attributed to some overfitting in more complex models, e.g., ARIMA(6,1,6) and ARIMA(7,1,6). The candidate model mechanism resolved such issues by increasing the accuracy and robustness.

According to Figure 9, the labels of the dataset were somewhat imbalanced. Thus, a confusion matrix analysis was conducted to check the possible existence of biased predictions. For brevity, only the best result of ARI(7,1) without OL is presented in Figure 12a. It is evident that high values were achieved on the diagonal cells, while low values were achieved on the off-diagonal ones, which indicates that the classifier predicted each class accurately without a large bias. Thus, for Task 1, the best pipeline selected in stage 2 (step 5 in Figure 1) was the EVC classifier using features extracted from ARI(7,1).

From the shaking table tests, Table 1, the natural period of the test structure varied from 0.56 s (UD) to 1.36 s (AL). As mentioned in Section 5.1, the floor information was not considered in Task 1 where the whole structure was treated as a SDOF system with damage detected as changes in its global stiffness and natural period, Figure 7. From the results, the ML classifiers were sensitive to the changes in the ARIMA coefficients corresponding to the changes in the first natural period. This is consistent with findings of high damage correlations with the AR coefficients of the ARMA models [2,21].

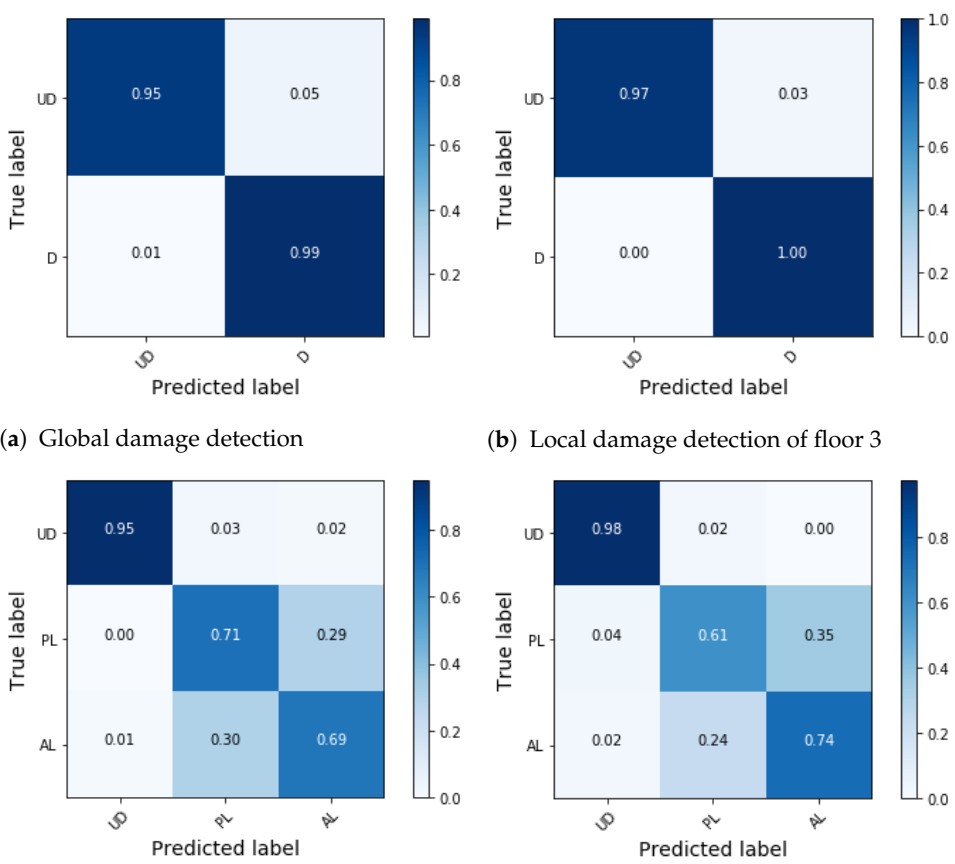

(**a**) Global damage detection                  (**b**) Local damage detection of floor 3

(**c**) Local damage pattern recognition of floor 1    (**d**) Local damage pattern recognition of floor 2

**Figure 12.** Confusion matrices of ARI(7,1) with EVC.

### 7.2. Task 2: Local Damage Detection

In Task 2, the local detection state was based on the floor level where only TS signals collected on the same floor were used for training and testing the ML models. From Table 6, the local damage detection achieved over 95% accuracy. Similar to the observations in Task 1, EVC achieved the highest accuracy compared to the individual classifiers in all cases, confirming the stable and robust performance of the EVC. At the floor level, using features extracted from the ARI(7,1) model still slightly outperformed the other two complex models especially for floors 1 and 2; however, all three models had the same performance for floor 3 with EVC (over 98% accuracy).

In addition, the example confusion matrix for floor 3, presented in Figure 12b, indicates the accurate and unbiased performance of EVC using ARI(7,1) features under a slightly imbalanced label distribution (41% UD vs. 59% D) in the local damage detection Figure 9. These promising results are attributed to more significant local stiffness changes of floor 3 due to brace loosening where the effective area of the braces for floor 3 was less than half that for the other floors, Figure 4a. Therefore, from these results, it is inferred that the ARIMA coefficients capture the local stiffness changes between floors giving more confidence for their potential in practical applications.

**Table 6.** Accuracy (%) of the local damage detection for different candidate models.

| Model | Floor # | SVM | NN | RF | LR | EVC |
|---|---|---|---|---|---|---|
| ARI(7,1) | 1 | 96.7 | 96.7 | 93.4 | 96.4 | 96.7 |
| | 2 | 97.5 | 97.2 | 96.2 | 97.4 | 97.7 |
| | 3 | 98.5 | 98.7 | 98.0 | 98.5 | 98.7 |
| ARIMA(6,1,6) | 1 | 95.3 | 95.7 | 88.4 | 93.4 | 95.8 |
| | 2 | 96.2 | 96.0 | 92.3 | 94.5 | 96.4 |
| | 3 | 97.8 | 97.6 | 93.9 | 97.6 | 98.0 |
| ARIMA(7,1,6) | 1 | 95.8 | 95.3 | 88.7 | 92.9 | 95.8 |
| | 2 | 96.9 | 95.9 | 91.9 | 93.5 | 96.9 |
| | 3 | 97.6 | 97.8 | 94.1 | 97.0 | 98.0 |

*7.3. Task 3: Local Damage Pattern Recognition*

From the two validation experiments in Tasks 1 and 2, it was shown that the ARIMA coefficients could accurately distinguish between damaged and undamaged states both globally and locally. However, in the present experiment with increased difficulty of the task, i.e., more classes to be classified, Table 7 shows that the performance of all models slightly decreased, especially in the complex models with RF and LR classifiers. In general, EVC provided the best accuracy of about 80% in all cases. ARI(7,1) with EVC again presented the best performance with the highest accuracy, i.e., 81.5% and 82.1% in floors 1 and 2, respectively. Although the results were lower than those in Tasks 1 and 2, they were much higher than a random guess of 33% for Task 3 three-class classification.

The confusion matrix analysis, Figure 12b–d, indicated that the best classifier was more accurate in distinguishing between undamaged and damaged states but less accurate to identity the differences between damage levels (PL and AL). Moreover, Figure 12c,d shows that the model was prone to predict PL as being AL for the braces. This misclassification is explained by a coupling effect from different floors, where the nonlinear response of a single floor is influenced by the adjacent floors [14].

Taking test record EQ12 as an example, there was only PL in floors 1 and 2 but AL in floor 3 during EQ12, Table 1. We inferred that the nonlinear response of floor 2 was influenced by both floor 3 with severe damage (AL) and floor 1 with a similar damage level (PL). Thus, using the ARIMA model with such simple structural idealization (each floor treated independently) may not render a very accurate damage pattern recognition task. Future work should make use of more accurate substructure idealization of the tested structural system [14].

**Table 7.** Accuracy (%) of the local damage pattern recognition for different candidate models.

| Model | Floor # | SVM | NN | RF | LR | EVC |
|---|---|---|---|---|---|---|
| ARI(7,1) | 1 | 80.8 | 80.2 | 80.4 | 79.2 | 81.5 |
| | 2 | 80.7 | 81.2 | 79.6 | 76.3 | 82.1 |
| ARIMA(6,1,6) | 1 | 79.4 | 79.2 | 69.5 | 78.1 | 80.8 |
| | 2 | 79.5 | 80.0 | 72.0 | 70.4 | 80.5 |
| ARIMA(7,1,6) | 1 | 78.9 | 78.3 | 72.8 | 80.0 | 80.0 |
| | 2 | 79.5 | 79.1 | 72.5 | 70.0 | 79.8 |

Further evaluation of the the test accuracy of floor damage pattern from a holistic point of view for a single event, i.e., predicting labels for a certain earthquake, can be based on majority labels predicted among the TS segments of each floor. Subsequently, normalizing the predictions for each label, a probability distribution is obtained, and the final event prediction is based on the highest probability. The predictions from a single test earthquake using ARI(7,1) are presented in Figure 13, and the final predictions were consistent with the true labels in the Table 1.

Similar to the observations in Figure 12c,d, the model can easily distinguish between undamaged and damage states (EQ4 and EQ7) with high confidence. On the contrary, the difference between PL and AL was less significant (EQ12 and EQ15) but still acceptable. Therefore, the classification performance achieved in damage pattern recognition is also applicable in practice, especially from a holistic point of view for a single earthquake event regarding the typically difficult task of *damage pattern prediction*.

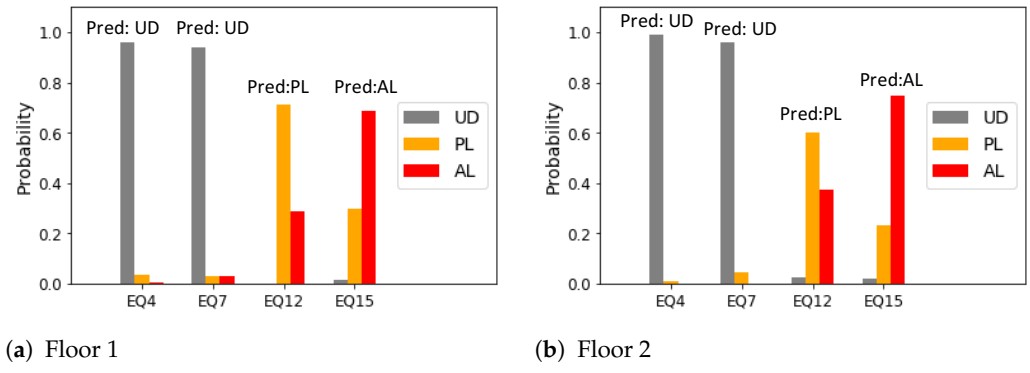

(**a**) Floor 1        (**b**) Floor 2

**Figure 13.** Damage pattern predictions of a single earthquake in test set using the ARI(7,1) model.

### 7.4. Feature Importance Analysis

Evaluating the feature importance of the RF classifier in the classification module revealed useful information regarding the feature selection for damage detection. This was shown to be reasonable since RF using features extracted from ARI(7,1) achieved an accuracy of 95% or higher in Tasks 1 and 2 and an acceptable accuracy of 80% in Task 3. Three cases were selected to report the *FI* score of each feature with respect to all features, Figure 14. In Figure 14a,b, the highest order coefficients, i.e., AR 7, had a dominating effect with the largest share of the *FI* score of about 0.4 for Tasks 1 and 2.

AR orders 6 and 1 had important contributions for the *FI* score in the range of 0.15 to 0.2. Considering the high accuracy (97–98%) achieved in these two tasks, we concluded that all AR orders contained essential information sensitive to damage detection, especially the highest AR order, e.g., 7, in our case. Thus, using all AR coefficients as features is recommended based on this study, instead of only using low-order coefficients, i.e., AR orders 1, 2, and 3 as suggested in [2,3].

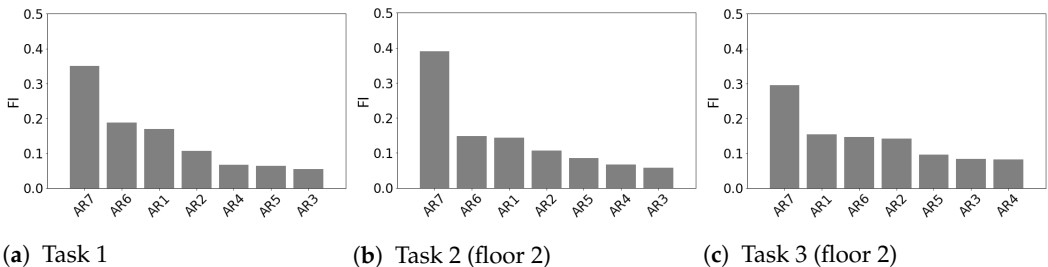

(**a**) Task 1          (**b**) Task 2 (floor 2)          (**c**) Task 3 (floor 2)

**Figure 14.** *FI* score of features extracted by the ARI(7,1) model.

In Task 3, compared with Tasks 1 and 2, the classification problem was expanded from binary to three-class (Figure 14b,c). We observed that high order contributions still existed, and the bar plot presents a similar trend to that of Tasks 1 and 2. However, the *FI* score of the AR 7 decreased from 0.38 to 0.29, and the *FI* score of the lower orders increased by 0.02∼0.03, especially for AR 2. This observation further supports the need to engage all extracted features, i.e., all AR coefficients.

## 8. Conclusions and Extensions

In this study, to overcome the constraints of stationary conditions in TS modeling and promote the automatic ML technology in SHM applications, we proposed a two-stage framework, namely ARIMA-ML. In stage 1, through the **pre-processing module**, the raw non-stationary TS data was processed to be stationary to satisfy the assumptions of the ARMA modeling. Through the **model parameter determination module**, multiple candidate models were selected to increase the robustness of the framework.

In stage 2, the segmented data were processed by the selected candidate model to obtain feature–label pairs in the **feature extraction module**. Subsequently, an ensemble classifier EVC engaged with multiple single classifiers was trained and tested in the **classification module**. This was followed by repeating stage 2 for all candidate models, such that the best model, from a performance accuracy point of view, was finally determined for use with future predictions.

The effectiveness of the ARIMA-ML framework on the damage recognition of building structures was validated using TS data (mostly non-stationary) from shaking table tests for a steel structure subjected to different earthquake hazard levels producing several scenarios of damage detection tasks. Three classification experiments were designed: (1) global damage detection, (2) local damage detection, and (3) local damage pattern recognition. The validation experimental results demonstrated the robustness and accurate performance of the ARIMA-ML in all tasks, where nearly 97%, 98%, and 80% average segment accuracy were obtained for these three tasks, respectively.

For the simpler binary Tasks 1 and 2, the confusion matrix analysis results further indicated the unbiased model performance even under an imbalanced-class distribution. Along with different experiments, discussions were made to explore the influence of overlapping pattern for data augmentation in the SSND procedure and the effect of higher order terms of the extracted features based on the *FI* scores to assess the feature performance. Our specific conclusions are listed as follows:

1. The ARIMA coefficients were sensitive to the changes in stiffness both globally and locally. Using these coefficients as features, the developed framework accurately distinguished between the undamaged and damaged states with accurate performance in the more fine-grained damage categories, i.e., multiple damage patterns representing different damage levels.
2. The overlapping pattern (OL) used in the sliding window did not significantly affect the results, which we attributed to the size of the dataset (over 7000 segments) generated by Non-OL. This dataset contained sufficient information to accurately train a classifier for global damage detection, and applying OL slightly improved the classification accuracy (0.1%∼0.3%). However, the OL increased the computational

cost, and a trade-off between efficiency and accuracy should be considered in different practical applications involving real-time SHM field implementations.

3. Compared to previous studies, the candidate model mechanism made the framework more robust. In our case, the simpler TS model, ARI($p, I$), achieved better classification performance compared with the other complex ones, ARIMA($p, I, q$), where $p = 6$ or $7$, $I = 1$, and $q = 6$. This was mainly due to some overfitting in these more complex models, which degraded the classification performance. However, the complex models were shown to have better fitting results through the AIC value evaluation and residual diagnosis. Thus, considering several candidate models instead of using one best fitted model can reduce the risk of overfitting.

4. Due to the good performance of RF, its important characteristic of feature importance (*FI*) score was evaluated to understand how each feature was weighted using this ML classifier for making the final decision. The *FI* scores indicated the influence of the higher order AR coefficients, e.g., AR 7 and 6, and some lower order coefficients, e.g., AR 1 and 2. Therefore, it is important to engage *all ARIMA coefficients* as damage features for the accurate and stable performance of the proposed ARIMA-ML framework.

Even though the proposed ARIMA-ML framework achieved promising results in the studied experiments, there still exist certain drawbacks that require future study. These drawbacks and the corresponding future suggestions are summarized as follows:

1. The presented framework was validated using limited shaking table tests of a TCBBF, where the structural type and damage scenarios were not sufficient due to the high expense in conducting other full-scale experiments. More validations using physical tests or detailed finite element analyses are needed in the future. Moreover, using transfer learning techniques [40] to only train on simulated data but with fine-tuning and tests from real experimental data has high potential for more efficient and practical research directions related to ML adoption in SHM.

2. The TS models and ML classifiers need to fit the available data with the possible risk of underfitting or overfitting, e.g., when using inappropriate orders of the ARIMA model or poorly tuned parameters in the ML classifiers. Even though selecting several candidate models instead of using one single "best" model provides robustness of the framework, this is a subjective and costly approach, which requires further investigation.

3. Substructure idealization is suggested for the improvement in damage localization tasks of framed structures, where more advanced structural idealization techniques beyond SDOF and MDOF approaches, Figure 7, can be explored.

**Author Contributions:** Conceptualization and methodology, Y.G., K.M.M., and Y.C. (Yueshi Chen); Software and validation, Y.G. and K.M.M.; Formal analysis and interpretation, Y.G., K.M.M., and Y.C. (Yueshi Chen); Data curation, Y.G., Y.C. (Yueshi Chen), W.W., and Y.C. (Yiyi Chen); Original draft preparation, Y.G., K.M.M., and Y.C. (Yueshi Chen); Supervision and final revision, K.M.M.; Funding acquisition, K.M.M., W.W., and Y.C. (Yiyi Chen). All authors have read and agreed to the published version of the manuscript.

**Funding:** This research received funding support from Tsinghua-Berkeley Shenzhen Institute (TBSI), China; California Department of Conservation, California Geological Survey, Strong Motion Instrumentation Program agreement 1019-016; Taisei Chair of Civil Engineering, University of California, Berkeley; and California Department of Transportation (Caltarns) for "Bridge Rapid Assessment Center for Extreme Events (BRACE2)" project (TO 001), part of the PEER–Bridge Program agreement 65A0774. The shaking table test was funded by the Natural Science Foundation of China (Grant No. 51578403 and 51820105013).

**Institutional Review Board Statement:** Not applicable.

**Informed Consent Statement:** Not applicable.

**Data Availability Statement:** Some or all data, models, or code that support the findings of this study are available from the authors upon reasonable request.

**Conflicts of Interest:** The authors declare no conflict of interest.

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
