# Peer review of "Auto-Regressive Integrated Moving-Average Machine Learning for Damage Identification of Steel Frames"

_applsci, doi:10.3390/app11136084_

Round 1
Reviewer 1 Report
- Chapter 5 is titled "Numerical Validation Experiments". There's no obvious numerical procedure such as simulations described to which the term "numerical" would apply.
- Line 176-177: How is the highest possible value determined?
- Line 255: Rephrase to "less cost efficient" to keep in line with scientific language
- Line 271: There is an "if" clause introduced in the sentence, but there is no subordinate clause with causal relation to it.
- Line 300: Please add a reference to what or who is suggesting higher order models.
- Figure 3: (Pseudo-Code) For better readability, please add vertical space between the second thick line, that seperates "Set j =..." from the header.
- Line 326: Please add the abbreviation "UD" since it's mentioned in Table 1.
- Figure 4a: Please change the line colour and type of the yellow arrows to distinguish them from the braces.
- Figure 4b:Units missing
- Line 359-362: For better readability and to ease understanding, please rephrase the sentence.
- Line 368-370: For better readability and to ease understanding, please rephrase the sentence.
- Line 377: How does the number of measurements 13 (13x3x3) come about?
- Line 526-529: For better readability and to ease understanding, please rephrase the sentence.
- Line 547: indefinite article missing "..an accuracy of..." and "an acceptable accuracy..."
- Line 526-569: For better readability and to ease understanding, please rephrase the sentence.
- Line 584: Please replace or remove the vague term "very"
- Line 594-596: For better readability and to ease understanding, please rephrase the sentence. It's not clear whether ARI performs better, worse or similar to ARIMA.
- Line 608, 614, 619: Please choose a consistent style for bullet caharacters, as they are different from those used previously.

Author Response
Please see the attachment.
--
We would like to thank the reviewers for the precious time to review our paper and list the comments and suggestions to improve our paper. We believe that addressing the reviewers’ remarks has made the article better and clearer. The changes to the manuscript are highlighted in red in the revised manuscript file. Please, find below the point-by-point answers to the reviewers’ comments.
Reviewer #1
- Chapter 5 is titled "Numerical Validation Experiments". There's no obvious numerical procedure such as simulations described to which the term "numerical" would apply.
Reply: We agreed with the reviewer’s comments and modified the chapter title as “Validation Experiments”.
- Line 176-177: How is the highest possible value determined?
Reply: The highest average segments accuracy is determined from results computed from each pair. We modified the sentences as follows:
“The candidate models are evaluated by average segments accuracy (ratio between the number of correct predicted segments and the total number of the input segments). Stage 2 is repeated for each candidate pair, and the best model is determined with the highest average segments accuracy”
- Line 255: Rephrase to "less cost efficient" to keep in line with scientific language
Reply: We agreed with the change and modified the sentence.
- Line 271: There is an "if" clause introduced in the sentence, but there is no subordinate clause with causal relation to it.
Reply: The subordinate is “all coefficients form a p-dimensional feature vector”.
- Line 300: Please add a reference to what or who is suggesting higher order models.
Reply: Added the reference (Shumway & Stoffer, 2017).
- Figure 3: (Pseudo-Code) For better readability, please add vertical space between the
second thick line, that separates "Set j =..." from the header.
Reply: Fixed.
- Line 326: Please add the abbreviation "UD" since it's mentioned in Table 1.
Reply: Added. “If no damage pattern is observed, it is recorded as undamaged (UD).”
- Figure 4a: Please change the line colour and type of the yellow arrows to distinguish them from the braces.
Reply: Updated.
- Figure 4b: Units missing
Reply: Added in the sub-figure caption.
- Line 359-362: For better readability and to ease understanding, please rephrase the sentence.
Reply: We rephrased the sentences. “Besides, the importance of each coefficient was further evaluated using feature importance () score. While the RF classifier is performing classification, the fed-in data are split into multiple subsets, where each subset belongs to one specific class (e.g., UD, PL, or AL in our case). The split criteria are based on features and their weights. The more important features make the subset able to distinguish the class better, and subsequently the corresponding weights are used to compute the score.”
- Line 368-370: For better readability and to ease understanding, please rephrase the sentence.
Reply: We modified the sentences as follows: “In the pre-processing module, according to the sampling frequency (256 Hz) of the sensors used in the shaking table experiments, the segmentation window size was taken as 200 data points. Thus, the duration of each segment is about 0.78 second , which was considered sufficient in this study.”
- Line 377: How does the number of measurements 13 (13x3x3) come about?
Reply: Added more details in the sentence: “a total of 117 (13 records × 3 floors × 3 sensor locations)”
- Line 526-529: For better readability and to ease understanding, please rephrase the sentence.
Reply: We rephrased the sentences. “This misclassification is explained by a coupling effect from different floors, where the non-linear response of a single floor is influenced by the adjacent floors (Xing & Mita 2012). Taking test record EQ12 as an example, there was only PL in floors 1 and 2 but AL in floor 3 during EQ12, Table 1. It is inferred that the nonlinear response of floor 2 was influenced by both floor 3 with severe damage (AL) and floor 1 with a similar damage level (PL).”
- Line 547: indefinite article missing "..an accuracy of..." and "an acceptable accuracy..."
Reply: Fixed.
- Line 526-569: For better readability and to ease understanding, please rephrase the sentence.
Reply: We rephrased the sentences. “In this study, to overcome the constraints of stationary conditions in TS modeling and promote the automatic ML technology in SHM applications, we proposed a two-stage framework, namely ARIMA-ML. In stage 1, through the pre-processing module, the raw non-stationary TS data can be processed to be stationary to satisfy the assumptions of the ARMA modeling. Through the model parameter determination module, multiple candidate models are selected to increase the robustness of the framework. In stage 2, the segmented data are processed by the selected candidate model to obtain feature-label pairs in the feature extraction module. Subsequently, an ensemble classifier EVC engaged with multiple single classifiers is trained and tested in the classification module. This is followed by repeating stage 2 for all candidate models, such that the best model, from a performance accuracy point of view, is finally determined, which is to be used for future predictions.”
- Line 584: Please replace or remove the vague term "very"
Reply: Removed.
- Line 594-596: For better readability and to ease understanding, please rephrase the sentence. It's not clear whether ARI performs better, worse or similar to ARIMA.
Reply: We rephrased the sentences. “In our case, the simpler TS model, ARI(), achieved better classification performance than other complex ones, ARIMA(), where or 7, , and . This is mainly due to some overfitting in these more complex models, which degrades the classification performance.”
- Line 608, 614, 619: Please choose a consistent style for bullet caharacters, as they are different from those used previously.
Reply: Fixed.
Reviewer 2 Report
The Authors should be aware of some contributions which have been published in the scientific Literature and are relevant to the topic they dealt with in the paper they submitted for publication in to the journal "Applied Sciences", namely:
- Andreaus U., Baragatti P., Fatigue crack growth, free vibrations and breathing crack detection of Aluminium Alloy and Steel beams. J. of Strain Analysis for Engineering Design, 2009, 44(7); p. 595-608. doi: 10.1243/03093247JSA527
- Andreaus U., Baragatti P., Experimental damage detection of cracked beams by using nonlinear characteristics of forced response. Mech. Syst. Signal Process. 2012, 31(8), 382- 404. DOI: 10.1016/j.ymssp.2012.04.007
- Andreaus U., Casini P., Identification of multiple open and fatigue cracks in beam-like structures using wavelets on deflection signals, Continuum Mechanics and Thermodynamics, First online: 19 May 2015, Vol. 28(1-2), 1 March 2016, pp. 361-378, DOI: 10.1007/s00161-015-0435-4.
The paper is interesting and falls within the topics dealt with by "Applied Sciences". Thus, I recommend its publication in "Applied Sciences", provided that all the raised issues will be properly addressed, namely, the above mentioned papers will be conveniently cited in the revised version of the submitted paper.
Author Response
Please see the attachment.
--
- The Authors should be aware of some contributions which have been published in the scientific Literature and are relevant to the topic they dealt with in the paper they submitted for publication in to the journal "Applied Sciences", namely:
- Andreaus U., Baragatti P., Fatigue crack growth, free vibrations and breathing crack detection of Aluminium Alloy and Steel beams. J. of Strain Analysis for Engineering Design, 2009, 44(7); p. 595-608. doi: 10.1243/03093247JSA527
- Andreaus U., Baragatti P., Experimental damage detection of cracked beams by using nonlinear characteristics of forced response. Mech. Syst. Signal Process. 2012, 31(8), 382- 404. DOI: 10.1016/j.ymssp.2012.04.007
- Andreaus U., Casini P., Identification of multiple open and fatigue cracks in beam-like structures using wavelets on deflection signals, Continuum Mechanics and Thermodynamics, First online: 19 May 2015, Vol. 28(1-2), 1 March 2016, pp. 361-378, DOI: 10.1007/s00161-015-0435-4.
Reply: We thank to the reviewer’s suggestions and added the listed references in the revised manuscripts.